# Synthesis, Biological Evaluation, and In Silico Studies of New Acetylcholinesterase Inhibitors Based on Quinoxaline Scaffold

**DOI:** 10.3390/molecules26164895

**Published:** 2021-08-12

**Authors:** Paptawan Suwanhom, Jirakrit Saetang, Pasarat Khongkow, Teerapat Nualnoi, Varomyalin Tipmanee, Luelak Lomlim

**Affiliations:** 1Department of Pharmaceutical Chemistry, Faculty of Pharmaceutical Sciences, Prince of Songkla University, Hat Yai 90112, Songkhla, Thailand; 6010730017@email.psu.ac.th; 2Phytomedicine and Pharmaceutical Biotechnology Excellent Center (PPBEC), Faculty of Pharmaceutical Sciences, Prince of Songkla University, Hat Yai 90112, Songkhla, Thailand; 3Department of Surgery, Faculty of Medicine, Prince of Songkla University, Hat Yai 90112, Songkhla, Thailand; 5510330008@email.psu.ac.th; 4Department of Biomedical Sciences and Biomedical Engineering, Faculty of Medicine, Prince of Songkla University, Hat Yai 90112, Songkhla, Thailand; pasarat.k@psu.ac.th; 5Department of Pharmaceutical Technology, Faculty of Pharmaceutical Sciences, Prince of Songkla University, Hat Yai 90112, Songkhla, Thailand; teerapat.n@psu.ac.th

**Keywords:** quinoxaline, acetylcholinesterase inhibitors, butyrylcholinesterase inhibitors, ADME prediction, molecular docking, cytotoxicity

## Abstract

A quinoxaline scaffold exhibits various bioactivities in pharmacotherapeutic interests. In this research, twelve quinoxaline derivatives were synthesized and evaluated as new acetylcholinesterase inhibitors. We found all compounds showed potent inhibitory activity against acetylcholinesterase (AChE) with IC_50_ values of 0.077 to 50.080 µM, along with promising predicted drug-likeness and blood–brain barrier (BBB) permeation. In addition, potent butyrylcholinesterase (BChE) inhibitory activity with IC_50_ values of 14.91 to 60.95 µM was observed in some compounds. Enzyme kinetic study revealed the most potent compound (**6c)** as a mixed-type AChE inhibitor. No cytotoxicity from the quinoxaline derivatives was noticed in the human neuroblastoma cell line (SHSY5Y). In silico study suggested the compounds preferred the peripheral anionic site (PAS) to the catalytic anionic site (CAS), which was different from AChE inhibitors (tacrine and galanthamine). We had proposed the molecular design guided for quinoxaline derivatives targeting the PAS site. Therefore, the quinoxaline derivatives could offer the lead for the newly developed candidate as potential acetylcholinesterase inhibitors.

## 1. Introduction

Dementia is defined as a group of progressive neurological disorders that deteriorate memory, thinking, behavior and emotion. Dementia can occur in diverse forms, including Alzheimer’s disease (AD), vascular dementia, dementia with Lewy bodies and frontotemporal dementia. As Alzheimer’s disease contributes almost 70 percent of the cases, especially in older adults, it is considered the major form of dementia. In 2020, Alzheimer’s Disease International (ADI) reported over 50 million cases of dementia worldwide. With over 10 million new cases globally each year, the number of dementia cases was anticipated to rise to 131.5 million by 2050 [1].

Three major hypotheses have been suggested to describe the pathogenesis and, in turn, provide molecular targets for the discovery of effective agents for Alzheimer’s disease pharmacotherapy. Histopathological characteristics of AD include extracellular beta-amyloid (Aβ) plaques and intracellular aggregations of neurofibrillary tangles (NFTs). Aβ plaque resulted from the accumulation of insoluble Aβ40 fragments, a product of abnormal cleavage of amyloid precursor protein (APP). This leads to the beta-amyloid hypothesis for the pathogenesis of AD. The Tau hypothesis stems from hyperphosphorylation of the microtubule-stabilizing tau protein, which induces tau protein oligomerization and dissociation of tubule subunits. Aggregation of the phosphorylated tau filaments results in neurofibrillary tangles, which are straight, fibrillar and highly insoluble patches in the neuronal cytoplasm. Amyloid plaques and NFTs play a role in the initiation and progression of neuronal damage and subsequent neuronal death [2,3]. The cholinergic hypothesis is based on selective loss of cholinergic neurons in the basal forebrain of AD patients. The deficit of presynaptic acetylcholine production led to cognitive impairment. Administration of inhibitors to acetylcholinesterase improved the symptoms of age-related cholinergic cognitive dysfunction [4,5].

However, clinical use of acetylcholinesterase inhibitors can only temporarily and partially relieve symptoms of Alzheimer’s disease, while the use of these agents could lead to undesirable side effects such as nausea, vomiting, diarrhea and dizziness. Acetylcholinesterase inhibitors were recently indicated to be prone to increase the risk of depression in AD patients [6,7]. Alzheimer’s disease etiology is multifaceted. Besides the three major hypotheses, other features such as oxidative stress [8], biometal ions accumulation [9], and neuroinflammation [10] also participate in AD pathogenesis. Therefore, using an agent that can work simultaneously on several targets associated with AD pathogenesis—the multi-target-directed ligand (MTDL) has been suggested [11,12,13].

According to the cholinergic hypothesis of AD pathogenesis, modulation of cholinergic neurotransmission by acetylcholinesterase inhibitors has been the mainstay for AD pharmacotherapy for decades [14]. Acetylcholine is a neurotransmitter found in cholinergic neurons both in the central and peripheral nervous systems. In the central nervous system (CNS), this neurotransmitter is essential for learning and cognition. Acetylcholinesterase (AChE, E.C. 3.1.1.7) is a serine protease found at cholinergic postsynaptic neurons. This enzyme efficiently catalyzes the hydrolysis of acetylcholine and terminates neuronal transmission and signaling between cholinergic synapses [15]. Acetylcholinesterase inhibitors inhibit the function of AChE and subsequently result in acetylcholine accumulation. Butyrylcholinesterase (BChE, E.C. 3.1.1.8) is another member of the cholinesterase subfamily. In healthy brains, AChE hydrolyzes the major acetylcholine, while BChE only plays a secondary role. There is growing evidence that both AChE and BChE may be important in the development and progression of AD. Up to 45% of AChE may be lost in certain brain regions during the progression of this disease, while the levels of BChE activity conversely increases by up to 90% [16].

Tacrine was the first AChE inhibitor approved by The United States Food and Drug Administration (US FDA) since 1993 but later withdrawn from the market in 2013 due to its hepatotoxicity [17]. X-ray crystallography of tacrine–*Torpedo califonica* (*Tc*AChE) complex revealed strong interactions between tacrine and the Catalytic Anionic Site (CAS) at the bottom of deep and narrow AChE active site gorge [18,19]. Many compounds were reported as the possible candidate/lead for AChE inhibitors such as coumarins [20], flavonoids derivatives [20], flavonolignans [21], chalcones [22], quinolines [23], quinoxalines [24,25] and others. Among the possible leads, quinoxaline has been used as a molecular scaffold for the design of some acetylcholinesterase inhibitors [24,25,26,27,28,29,30]. 2-Phenylquinoxaline derivatives showed no inhibitory activity against AChE but showed moderate inhibitory activity against butyrylcholinesterase (BChE) [31]. 6-Aminoquinoxaline derivatives showed a neuroprotective effect in dopaminergic neuron culture [32,33]. However, the molecular mode of action of the quinoxaline remains unclear as well as the understanding of structural modification of the compound to improve its efficacy against AChE was still lack. 

In this work, we further explore the structure-activity relationships of the quinoxaline derivatives by variation of substituents at 2-,3-, and 6-position. Predictive ADME properties of the quinoxaline derivatives were anticipated by the SwissADME calculator. The synthesized compounds were evaluated for AChE and BChE inhibitory activity. The mode of enzyme inhibition was determined by an enzyme kinetic study. Cytotoxicity against human SH-SY5Y neuroblastoma cells of selected compounds was evaluated using a sulforhodamine B (SRB) assay. The binding interactions and AChE binding characteristics between the inhibitor and AChE were observed via in silico molecular docking. Further, structure modification was predicted and suggested based on in silico approach.

## 2. Results and Discussion

### 2.1. Chemistry

Employing tacrine as a lead compound, we designed and synthesized new acetylcholinesterase inhibitors based on quinoxaline scaffolds. Quinoxaline derivatives (**3a**–**5c**) were synthesized via the liquid-assisted grinding (LAG) method from *o*-phenylenediamine derivatives and glyoxal derivatives in ethanol as a solvent. The desired products were obtained in good yields (70–92%). The 6-amino quinoxalines (**6a**–**6c**) were prepared from the corresponding 2-nitro quinoxalines (**5a**–**5c**) via stannous (II) chloride reduction. The Chemical structure of the resulting compounds was confirmed by IR, ^1^H-NMR, ^13^C-NMR and HR-MS. The design strategy for the quinoxaline-based compounds and synthesis pathway is illustrated in Figure 1.

### 2.2. Predictive ADME Properties of Quinoxaline Derivatives

Physicochemical properties are crucial parameters for drug action in vivo. For the application as an anti-Alzheimer agent, an acetylcholinesterase inhibitor must be absorbed from the GI tract and be able to permeate blood brain barrier (BBB) to achieve its site of action in the central nervous system (CNS). The physicochemical properties of the synthesized compounds (**3a**–**6c**) were predicted by the SwissADME calculator based on the molecule’s lipophilicity, hydrogen bonding, rotatable bonds, topological polar surface area and compared with the drugs tacrine and galanthamine. The physicochemical descriptors and ADME properties of tested compounds **3a**–**6c** calculated by SwissADME are outlined in Table 1. All the synthesized compounds conformed to Lipinski’s rule of five and were predicted to have good GI absorption and be able to permeate BBB. These results suggested that the target compounds may have good pharmacokinetic properties.

### 2.3. Biological Evaluation

#### 2.3.1. Enzyme Inhibition Assay

The acetylcholinesterase and butyrylcholinesterase inhibitory activities of the quinoxaline derivatives (**3a**–**6c**) were determined by Ellman’s method [34] using human recombinant acetylcholinesterase (*Hu*AChE) and butyrylcholinesterase from equine serum (*Eq*BChE). Initially, acetylcholinesterase and butyrylcholinesterase inhibitory activities of the compounds were screened at the concentration of 100 µM. Compounds that showed higher than 50% inhibition at 100 µM were further evaluated for their half-maximal inhibitory concentration (IC_50_) values. IC_50_ values against AChE and BChE of the quinoxaline-based compounds are shown in Table 2. Percent inhibition at 100 µM of some inactive BChE inhibitors is shown in square brackets. The raw data of IC_50_ in this study were able to be accessed in Appendix A.

The synthesized compounds showed moderate to potent acetylcholinesterase inhibitory activity with IC_50_ values ranging from 50.08 to 0.077 µM. 2,3-Dimethylquinoxalin-6-amine (**6c**) (IC_50_ = 0.077 µM) exhibited the highest AChE inhibitor activity in this series with slightly higher potency than tacrine (IC_50_ = 0.11 µM) and galanthamine (IC_50_ = 0.59 µM). For butyrylcholinesterase inhibitory activity, compounds **3a**, **3b**, **5a** and **5b** showed only moderate potency with IC_50_ values ranging from 60.95 to 14.91 µM. Other compounds were considered inactive butyrylcholinesterase inhibitors. Compounds **3c**, **4a**–**4c**, **5c** and **6a**–**6c** were AChE selective while **3a**–**3b** and **5a**–**5b** did not show selectivity between AChE and BChE.

Quinoxaline (**3a**) showed moderate acetylcholinesterase inhibitory activity with an IC_50_ value of 13.22 µM. Introduction of 2-phenyl group (**3b)** resulted in diminished activity (IC_50_ = 50.08 µM). A substitution of 2,3-dimethyl resulted in higher potency (IC_50_ = 7.25 µM). Substitution of 6-position of **3a** with electron-withdrawing group such as 6-chloro (**4a**, IC_50_ = 23.87 µM) and 6-nitro (**5a**, IC_50_ = 21.31 µM) group led to minimized activity. This trend was also observed in the 2,3-dimethylquinoxaline derivatives. Compounds **4c** and **5c** showed diminished activity against AChE, but **6c** exhibited the most potent acetylcholinesterase inhibitory activity among all compounds in this series, with an IC_50_ value of 0.077 µM. The 2,3-dimethylquinoxalin-6-amine (**6c**) demonstrated higher potency than tacrine (IC_50_ = 0.107 µM) and galanthamine (IC_50_ = 0.59 µM), which are known AChE inhibitors. In contrast, 6-aminoquinoxaline (**6a**) with the electron-donating group at 6-position showed potent acetylcholinesterase activity with an IC_50_ value of 0.74 µM. Among the 2-phenylquinoxaline derivatives, the substitution of R with both electron-withdrawing (**4b**, **5b**) and electron-donating (**6b**) groups resulted in higher activity with IC_50_ values ranging from 1.31–39.0 µM.

The substitution of quinoxaline (**3a**, IC_50_ = 40.64 µM) with 2-phenyl group (**3b**, IC_50_ = 14.91 µM) resulted in elevated butyrylcholinesterase inhibitory activity. Derivatives with 2,3-dimethyl substitution (**3c**, **4c**, **5c** and **6c**, R_1_=R_2_=CH_3_) showed no activity against butyrylcholinesterase. Substitution of R with chloro-, nitro- and amino groups led to diminished or lack of butyrylcholinesterase inhibitory activity. Compounds **3a**, **3b**, **5a** and **5b** could inhibit both AChE and BChE.

In 2014, Zeb A. et al. reported 2-phenylquinoxaline analogues as selective butyrylcholinesterase inhibitors [31]. In that work, the structure of the quinoxaline scaffold was modified on the 2-phenyl group by substitution with electron-withdrawing or electron-donating groups. The resulted compounds showed moderate butyrylcholinesterase inhibitory activity with IC_50_ values ranging from 57.1 to 7.7 µM, but all of the 2-phenyl quinoxaline analogues were found to be inactive against acetylcholinesterase. In this work, we report for the first time the moderate to potent acetylcholinesterase inhibitory activity of quinoxaline analogues.

#### 2.3.2. Enzyme Kinetic Study

The kinetic studies for the most active inhibitor of Human recombinant AChE (**6c**) were performed to illustrate the involved mechanism of AChE inhibition. For this purpose, three fixed inhibitory concentrations of test compound **6c** (30, 50, 60 µM) were used, and for each concentration, the velocity (V) of the substrate was measured in the range of 10–250 µM. The Lineweaver–Burk plot (Figure 2) shows unchanged in V_max_ (56.59 µmol L^−1^S^−1^) and K_m_ (267.7 µmol L^−1^) as the concentration of the inhibitor increased, which reflected the mixed-type inhibition. In this case, the value of Alpha (0.1875) is very small (but greater than zero), and the inhibitor is bound almost entirely to the enzyme–substrate complex, and the mixed-type inhibition approaches an uncompetitive model.

#### 2.3.3. Cytotoxicity Evaluation

Selected compounds in the series were then evaluated for their cytotoxicity against SH-SY5Y neuron cells. As shown in Figure 3, results revealed that all the test compounds, **6a**, **6b** and **6c,** did not show significant toxicity at any tested concentrations to the neuron cells (LC_50_  >  100 µM) when compared with control. This result suggested the safety of the compounds when used with the neuronal cells.

### 2.4. In Silico Analysis of HuAChE Binding Characteristics

As the compounds **6a**, **6b** and **6c** exhibited the AChE potency, these compounds were selected to investigate more details in compound binding with AChE. The initial structure of *Hu*AChE was retrieved from the Protein Data Bank with PDB ID: 4EY5 [35,36]. A molecular modeling study was performed using AutoDock Tools-1.5.6, Discovery Studio 2021 Client [37] and Visual Molecular Dynamic (VMD) package. The structure of *Hu*AChE has its actives site at the bottom of a deep, narrow gorge (20 Å deep) composed of several major domains [38,39]. The peripheral anionic site (PAS) is located around the entrance of the active site gorge (approximately 18 Å away from the active site). This area consists of five amino acid residues, which are Tyr72, Asp74, Tyr124, Trp286 and Tyr341. The catalytic triad (CT) that is accountable for hydrolyzing the ester bond in acetylcholine consists of Ser203, His447 and Glu334 residues. The catalytic anionic site (CAS) in the neighborhood of the CT comprises Trp86, Tyr133, Tyr337, and Phe338. The acyl pocket which binds to the acyl group of acetylcholine consists of Phe295 and Phe297. In addition, the oxyanion hole that binds with the acetylcholine carbonyl oxygen, which has Gly121, Gly122 and Ala204, is shown in Figure 4.

In this study, first of all, we performed molecular docking with the large grid size (120 × 120 × 120 Å^3^) as the blind docking because the binding site of the synthesized quinoxaline derivative to *Hu*AChE had been unknown. The objective of this process was to roughly guide the possible pocket for the compounds. In Table 3, compounds **6a**, **6b** and **6c** were selected as representatives for the docking study. The molecular docking predicted that these compounds predictively bound the PAS site of *Hu*AChE, Figure 5. As the blind docking suggested that the synthesized compounds bound the PAS site, the PAS site-specific docking was again performed to visualize interactions between the compound and the site.

The peripheral anionic site of acetylcholinesterase lies at the entrance to the active site gorge. It is composed of five key residues Tyr 70 (72), Asp 72 (74), Tyr 121 (124), Trp 279 (286) and Tyr 334 (341); (*Torpedo californica* (*Tc*) numbering is given first, followed by mammalian numbering in brackets) [38]. From the molecular docking study, in Figure 5a, the Thioflavin T–*Hu*AChE complex showed that benzothiazol ring interacted with Trp286 via a π–π interaction and benzyl group displayed a π–π interaction with Trp286 and Tyr341, which bound equivalently at the PAS site of a *Tc*AChE [38]. Similarly, the docked poses of the synthesized compounds were illustrated in Figure 5b–d, respectively. Compound **6a** showed that the quinoxaline ring interacted with Trp286 and Phe338 with π–π interaction and exhibited hydrogen bonding with Asp74 and Arg296. For compound **6b**, the pyrazine ring of quinoxaline was observed to bind to PAS via a π–π interaction with Trp286 and Phe338. The nitrogen atom of the pyrazine ring showed a hydrogen bond with Arg296. The amino group that interacted with Ser293 showed a hydrogen bond, and the benzyl group displayed a hydrophobic interaction with Tyr337. Finally, compound **6c** used the quinoxaline ring to form a π–π interaction with Trp286 and Phe338. The nitrogen atom of the pyrazine ring showed a hydrogen bond with Arg296, and the methyl group displayed hydrogen interaction with Ser293. The amino group interacted with Asp74 via hydrogen bond. In summary, from the prediction, the hydrogen bond due to the amino group and π–π stacking due to quinoxaline led to the preference of the compounds towards the PAS site of the *Hu*AChE.

The mentioned result showed that quinoxaline derivatives (**6a**, **6b** and **6c**) bound the PAS site. This finding complied with the enzyme kinetic study that suggested **6c** as a mixed-type inhibitor that approached uncompetitive AChE inhibition. This meant the inhibitor tended to bind with the enzyme–substrate complex at another site. We then computationally designed additional quinoxaline derivatives so that the binding affinity could be improved. Herein, we considered the design guide based on the molecular docked pose of the quinoxaline derivatives (**6a**, **6b** and **6c**) with respect to thioflavin T due to the fact that thioflavin T is an AChE ligand bound at the PAS site [40,41]. The blockade at the PAS site can hinder the AChE-induced aggregation of beta amyloid [42,43]. The difference in structure surface area of the quinoxaline compound **6a** and thioflavin T (red circle) was visualized in Figure 6. The empty space would thus further explore an opportunity for the structure modification by replacing/adding different substituents, which could provide more interactions through hydrogen bonding with the polar nearby amino acids at the PAS site.

### 2.5. Hit-to-Lead Optimization of 6-Aminoquinoxaline Derivatives for PAS Site

Based on the docking of thioflavin T at the PAS site, we computationally designed the 6-aminoquinoxaline derivatives by adding a substituent in the quinoxaline ring. The added substituent was considered by the criterion that the substituent could provide hydrogen bonding and/or hydrophobic interactions with Tyr, Phe, and Trp. Besides, the predictive ADME properties were evaluated to justify whether this derivative could be attractive for next step development. The designed derivatives and ADME prediction were shown in Table 4 and Table 5, respectively. From Table 4, different alkyl halide, aniline, hydroxy group and others as substituents were introduced into the structure, and their affinity towards *Hu*AChE was assessed using molecular docking. Moreover, we design X group substitution at the ortho position and Y group substitution at the meta position on the 6-aminoquinoxaline.

In Table 4, compound **21** showed the binding energy of −7.13 kcal/mol, which has a methylacetamide group. In contrast, compound **27** displayed the highest binding energy of −4.80 kcal/mol, which has a carboxylic group. The conformers of compounds **21** and **27** were then selected to investigate the interaction at the PAS binding site of *Hu*AChE, Figure 7, along with thioflavin T. From the docked pose, compound **21**, Figure 7a, interacted with Tyr72 using the carbonyl oxygen. The quinoxaline ring showed π–π stack interaction with Trp286. Additionally, a nitrogen atom of pyrazine ring formed a hydrogen bond with Arg296. For compound **27** structure, the quinoxaline ring also showed π–π stack interaction with Trp286, while an amino group on the quinoxaline ring exhibited hydrogen bond with Ser293, Figure 7b. All structures of docked compounds in this study and interaction scheme between the compound and PAS site were available as Appendix A.

The predicted ADME parameters are presented in Table 5. All the new compounds (**7**–**36**) conformed to Lipinski’s rule of five and were predicted to have good GI absorption. In addition to their efficacy and no toxicity, the compounds intended for the treatment of AD must be able to cross the BBB to reach their target site. In general, lipophilicity was regarded to be the most important property, and its increased value often results in an improved in vitro activity [44,45]. According to Table 5, the compounds **8**–**11**, **15**–**18**, **23**–**26** and **30**–**33** exhibited permeate BBB.

To summarize, the quinoxaline ring played a major role in π–π stacking with Trp286. The hydrogen bonding due to the substituent group from the structure could also contribute as a hydrogen bond donor/acceptor, such as carbonyl, carboxyl or amine groups. We have speculated that polar substituent groups such as halogen could enhance the binding affinity to the PAS site of *Hu*AChE. Some carbonyl-related functional groups also could facilitate the enzyme binding from both ortho- and meta positions to amino groups in aromatic rings.

## 3. Materials and Methods

### 3.1. Chemistry

All chemicals used in the synthesis were purchased either from Sigma-Aldrich (St. Louis, MO, USA) or Merck AG (Darmstadt, Germany). The progress of synthesis reactions and the purities of the compounds were observed by thin-layer chromatography (TLC) on silica gel 60 F_254_ aluminum sheets obtained from Merck AG. Melting points were recorded using the Mel-TEMP II, LABORATORY DEVICES, USA. IR spectroscopy was performed on a Perkin Elmer spectrum, and principal absorptions were given in cm^−1^. ^1^H-NMR and ^13^C-NMR spectra were recorded by a BRUKER/AVANCETM NEO using deuterated chloroform (CDCl_3_) or dimethylsulfoxide (d_6_-DMSO) as solvent. In the NMR spectra, splitting patterns were designated as follows: s: singlet; d: doublet; t: triplet; m: multiplet. Coupling constants (*J*) were reported as Hertz. ESI-MS spectra were recorded on a Thermo Finnigan MAT 95XL. The Power Wave X, Biotele was used as a microplate reader.

### 3.2. Synthesis of Quinoxaline Derivatives (**3a**–**5c**)

Quinoxaline derivatives (**3a**–**5c**) were synthesized via the liquid-assisted grinding (LAG) method modified from a previous report [46]. A mixture of 2.5 mmol of *o*-phenylenediamine (**1a**) or substituted-*o*-phenylenediamine (**1b**, **1c**) and 5 mmol of glyoxal (**2a**) or substituted glyoxal (**2b**, **2c**) were ground in ethanol at room temperature. The progress of the reaction was monitored by TLC at regular intervals. While the neat reaction took 10–30 min for complete condensation, the solvent was evaporated by a rotary evaporator. The product was purified by silica gel column chromatography using dichloromethane as mobile phase and recrystallized from CH_2_Cl_2_. The nuclear magnetic resonance spectra of the synthesized compounds were available as Appendix A.

*Quinoxaline **3a***. Pale yellow solid; yield 92%; mp 28–30 °C. IR (KBr): 3444.7, 2921.1, 1496.7, 1369.7, 756.3 cm^−1^. ^1^H-NMR (500 MHz, DMSO-d_6_): δ 8.94 (2H, d, *J* = 1.98 Hz, H2, H3), 8.09 (2H, m, H5, H8), 7.86 (2H, m, H6, H7). ^13^C-NMR (125 MHz, DMSO-d_6_): δ 146.17, 142.72, 130.65, 129.60. ESI-MS: (*m*/*z*, [M + H]^+^) (Calcd: 130.05. Found: 131.0600).

*2-Phenylquinoxaline **3b**.* White solid; yield 87%; mp 77–79 °C. IR (KBr) 3432.4, 2931.0, 1736.1, 1543.7, 761.2 cm^−1^. ^1^H-NMR (500 MHz, DMSO-d_6_): δ 9.58 (1H, s, H3), 8.33 (2H, m, H5, H8), 8.13 (2H, m, H6, H7), 7.86 (2H, dddd, *J* = 1.60, 6.91, 8.05, 19.21 Hz, H1′, H5′), 7.59 (3H, m, H2′, H3′, H4′). ^13^C-NMR (125 MHz, DMSO-d_6_): δ 151.15, 143.88, 141.57, 141.23, 136.19, 130.76, 130.55, 130.04, 129.35, 129.26, 128.99, 127.60. ESI-MS: (*m*/*z*, [M + H]^+^) (Calcd: 206.08. Found: 207.09).

*2,3-Dimethylquinoxaline **3c***. Pale yellow solid; yield 78%; mp 104–106 °C. IR (KBr) 3447.1, 2921.1, 1633.7, 1398.8, 761.2 cm^−1^. ^1^H-NMR (500 MHz, DMSO-d_6_): δ 7.99–7.93 (2H, m, H5, H8), 7.74–7.70 (2H, m, H6, H7), 2.67 (6H, d, *J* = 1.30 Hz, 2xCH_3_). ^13^C-NMR (125 MHz, DMSO-d_6_): δ 154.45, 140.88, 129.24, 128.47, 23.22. ESI-MS: (*m*/*z*, [M + H]^+^) (Calcd: 158.08. Found: 159.09).

*6-Chloroquinoxaline **4a***. Pale yellow solid; yield 86%; mp 64–67 °C. IR (KBr) 3449.7, 2965.5, 1719.8, 1646.0, 1483.6, 1066.0, 801.3 cm^−1^. ^1^H-NMR (500 MHz, DMSO-d_6_): δ 8.98 (2H, dd, *J* = 1.82, 5.35 Hz, H2, H3), 8.19 (1H, d, *J* = 2.38 Hz, H8), 8.14 (1H, d, *J* = 8.95 Hz, H5), 7.90 (1H, m, H6). ^13^C-NMR (125 MHz, DMSO-d_6_): δ 147.20, 146.61, 142.99, 141.35, 135.02, 131.53, 131.34, 128.34. ESI-MS: (*m*/*z*, [M + H]^+^) (Calcd: 164.01. Found: 165.02).

*7-Chloro-2-phenylquinoxaline **4b***. White solid; yield 80%; mp 146–148 °C. IR (KBr) 3443.3, 2068.9, 1741.9, 1642.9, 1450.8, 1078.3, 687.7 cm^−1^. ^1^H-NMR (500 MHz, DMSO-d6): δ 9.60 (1H, s, H3), 8.33 (2H, dd, *J* = 1.74, 7.86 Hz, H5, H8), 8.21 (1H, d, *J* = 2.34 Hz, H6), 8.14 (1H, d, *J* = 8.90 Hz, H1′), 7.86 (1H, dd, *J* = 2.36, 8.90 Hz, H5′), 7.60 (3H, m, H2′, H3′, H4′). ^13^C-NMR (125 MHz, DMSO-d_6_): δ 152.36, 144.67, 142.31, 140.19, 136.08, 135.39, 131.23, 131.18, 130.89, 129.62, 128.32, 128.09. ESI-MS: (*m*/*z*, [M + H]^+^) (Calcd: 240.05. Found: 241.05).

*6-Chloro-2,3-dimethylquinoxaline **4c**.* White solid; yield 70%; mp 89–91 °C. IR (KBr) 3443.3, 2926.1, 1840.2, 1738.0, 1367.5, 1066.0, 830.1 cm^−1^. ^1^H-NMR (500 MHz, DMSO-d_6_): δ 8.72 (1H, d, *J* = 2.19 Hz, H8), 8.42 (1H, dd, *J* = 2.22, 9.06 Hz, H5), 8.15 (1H d, *J* = 9.08 Hz, H6), 2.74 (6H, d, *J* = 2.84 Hz, 2xCH_3_). ^13^C-NMR (125 MHz, DMSO-d_6_): δ 158.59, 157.55, 147.12, 143.63, 139.58, 130.36, 124.48, 122.72, 23.65. ESI-MS: (*m*/*z*, [M + H]^+^) (Calcd: 192.05. Found: 193.05).

*6-Nitroquinoxaline **5a***. Pale yellow solid; yield 89%; mp 174–176 °C. IR (KBr) 3445.9, 1737.4, 1525.6, 1351.3, 742.6 cm^−1^. ^1^H-NMR (500 MHz, DMSO-d_6_): δ 9.16 (2H, m, H2, H3), 8.92 (1H, m, H8), 8.57 (1H, dd, *J* = 2.63, 9.13 Hz, H6), 8.35 (1H, m, H5). ^13^C-NMR (125 MHz, DMSO-d_6_): δ 149.29, 148.56, 148.10, 145.12, 141.47, 131.71, 125.69, 123.99. ESI-MS: (*m*/*z*, [M + H]^+^) (Calcd: 175.04. Found: 176.04).

*7-Nitro-2-phenylquinoxaline **5b**.* Pale yellow solid; yield 87%; mp 185–187 °C. IR (KBr) 3462.9, 1738.0, 1524.2, 1351.8, 690.0 cm^−1^. ^1^H-NMR (500 MHz, DMSO-d_6_): δ 9.47 (1H, s, H8), 9.01 (1H, d, *J* = 2.53 Hz, H3), 8.54 (1H, dd, *J* = 2.5, 9.17 Hz, H6), 8.28–8.23 (3H, m, H1′, H5′), 7.60–7.58 (3H, m, H2′, H3′, H4′). ^13^C-NMR (125 MHz, DMSO-d_6_): δ 153.78, 147.35, 146.48, 146.37, 144.33, 139.93, 135.34, 131.30, 129.33, 128.26, 125.02, 123.99. ESI-MS: (*m*/*z*, [M + H]^+^) (Calcd: 251.07. Found: 252.07).

*2,3-Dimethyl-6-nitroquinoxaline **5c***. White solid; yield 79%; mp 133–135 °C. IR (KBr) 3447.1, 1737.6, 1523.1, 1346.4, 742.7 cm^−1^. ^1^H-NMR (500 MHz, DMSO-d_6_): δ 8.01 (1H, d, *J* = 2.06 Hz, H8), 7.97 (1H, d, *J* = 8.86 Hz, H6), 7.74 (1H, dd, *J* = 2.21, 8.85 Hz, H5), 2.68 (6H, d, *J* = 3.00 Hz, 2xCH_3_). ^13^C-NMR (125 MHz, DMSO-d_6_): δ 155.85, 155.16, 141.23, 139.50, 133.42, 130.36, 129.80, 127.26, 23.23. ESI-MS: (*m*/*z*, [M + H]^+^) (Calcd: 203.07. Found: 204.07).

### 3.3. Synthesis of 6-Aminoquinoxaline Derivatives (**6a**–**6c**)

The synthesis of 6-aminoquinoxaline derivatives (**6a**–**6c**) was adapted based on the previous study of quinoxaline derivatives with neuroprotective effect on dopaminergic neurons in Parkinson’s disease [32]. The mixture of substituted-6-nitroquinoxaline (**5a**–**5c**) (3.0 mmol), SnCl_2_ (26.4 mmol) and ethanol (25 mL) in a rounded-bottom flask equipped with a condenser was heated at reflux for 2 h. After the reaction mixture was cooled down to room temperature, 1M NaOH solution (20 mL) was added, and the desired product was extracted with ethyl acetate 50 mL 3 times. The product was purified by silica gel column chromatography using dichloromethane: methanol (80:20) as mobile phase and recrystallized from CH_2_Cl_2_. The nuclear magnetic resonance spectra of the synthesized compounds were available as Appendix A.

*Quinoxalin-6-amine **6a***. Yellow solid; yield 62%; mp 159–161 °C. IR (KBr) 3403.9, 1736.8, 1504.5, 1369.1, 1229.1, 860.3 cm^−1^. ^1^H-NMR (500 MHz, DMSO-d_6_): δ 8.58 (1H, d, *J* = 1.94 Hz, H2), 8.43 (1H, d, *J* = 1.92 Hz, H3), 7.71 (1H, d, *J* = 8.98 Hz, H5), 7.22 (1H, dd, *J* = 2.51, 9.03 Hz, H8), 6.90 (1H, d, *J* = 2.49 Hz, H6), 6.02 (2H, s, NH_2_). ^13^C-NMR (125 MHz, DMSO-d_6_): δ 181.32, 144.42, 139.14, 129.08, 121.84, 104.45. ESI-MS: (*m*/*z*, [M + H]^+^) (Calcd: 145.06. Found: 146.07).

*3-Phenylquinoxalin-6-amine **6b***. Yellow solid; yield 60%; mp 198–201 °C. IR (KBr) 3463.2, 1737.0, 1621.5, 1369.3, 1229.7, 694.9 cm^−1^. ^1^H-NMR (500 MHz, DMSO-d_6_): δ 9.23 (1H, s, H3), 8.18 (2H, dd, *J* = 1.68, 8.30 Hz, H1′, H5′), 7.88 (1H, d, *J* = 8.97 Hz, H5), 7.53 (2H, m, H2′, H3′), 7.46 (1H, m, H4′), 7.25 (1H, dd, *J* = 2.49, 8.99 Hz, H6), 6.95 (1H, d, *J* = 2.46 Hz, H8), 6.08 (2H, s, NH_2_). ^13^C-NMR (125 MHz, DMSO-d_6_): δ 150.64, 145.69, 143.77, 142.96, 137.04, 135.56, 129.96, 129.17, 129.06, 126.52, 122.89. ESI-MS: (*m*/*z*, [M + H]^+^) (Calcd: 221.10. Found: 222.10).

*2,3-Dimethylquinoxalin-6-amine****6c***. Brown solid; yield 65%; mp 186–188 °C. IR (KBr) 3448.2, 1736.1, 1641.7, 1369.4, 1217.0, 687.5 cm^−1^. ^1^H-NMR (500 MHz, DMSO-d_6_): δ 7.84 (1H, d, *J* = 9.07 Hz, H5), 7.36 (1H, dd, *J* = 2.17, 9.07 Hz, H8), 7.13 (1H, s, H6), 3.63 (2H, m, NH_2_), 2.73 (3H, s, CH_3_), 2.62 (3H, s, CH_3_). ^13^C-NMR (125 MHz, DMSO-d_6_): δ 150.97, 150.00, 148.47, 136.68, 129.63, 123.82, 21.74. ESI-MS: (*m*/*z*, [M + H]^+^) (Calcd: 173.10. Found: 174.10).

### 3.4. Evaluation of Acetyl- and Butyrylcholinesterase Inhibition

Human recombinant acetylcholinesterase (*Hu*AChE), butyrylcholinesterase from equine serum (*eq*BChE), acetylthiocholine iodide (ATCI), butyrylthiocholine iodide (BTCI) and 5,5′-dithiobis(2-nitrobenzoic acid) (DTNB) were gained from Sigma-Aldrich. Assays of acetyl- and butyrylcholinesterase inhibitory activity were performed according to Ellman’s method described previously [34]. Briefly, these assays were performed as follows. The reaction mixture (250 µL), containing 50 µL of buffer (50 mM Tris-HCl buffer (pH 8.0), 0.1 M NaCl, 0.02 M MgCl_2_.6H_2_O), 25 µL of 1.5 mM of ATCI or BTCI, 25 µL of 100 µM of test compounds in EtOH and 125 µL of 3 mM DTNB were added. Then, 25 µL of *Hu*AChE and equine serum of BChE in 50 mM Tris-HCl buffer containing 0.1% (*w*/*v*) BSA (pH 8.0). Reactions were initiated by the addition of the enzyme into the medium. This reaction resulted in the development of a yellow color, which was measured at 405 nm every 11s for 2 min in a Microplate Scanning Spectrophotometer. Each experiment was repeated in triplicate. In this study, tacrine and galanthamine were taken as reference drugs.

The percentage of enzyme inhibitory activity (%Inhibition) was calculated by the following expression: %Inhibition = ((Mean velocity of blank-Mean velocity of sample) × 100)/Mean velocity of blank. The IC_50_ value (the concentration of the compounds required for a 50% reduction in cholinesterase activity) was calculated using GraphPad Priam 2.01 software. Eight difference concentrations of the inhibitor (50 µM–1.6 × 10^−3^ µM) were used. The results are expressed as the mean ± SD. The selectivity of acetylcholinesterase inhibitory activity of the compounds can be evaluated by the ratio between IC_50_ of equine serum of BChE with IC_50_ of *Hu*AChE and shown as the Selectivity index (SI) [47].

### 3.5. Enzyme Kinetic Study

Kinetic studies of *Hu*AChE were performed by Ellman’s method as described above. In this study, the most potent AChE inhibitor (compound 6c) was selected for the determination of the AChE inhibitory mechanism. Kinetic characterization of the hydrolysis of ATCI for *Hu*AChE was carried out spectrometrically at 405 nm every 11 s for 2 min. A parallel control was run with the assay solution without inhibitor. The inhibition was evaluated by Lineweaver–Burk plot from substrate concentrations range between 10 to 250 µM for ATCI and inhibitor concentrations (0, 30, 50, 60 µM). The type of inhibition (competitive, uncompetitive, noncompetitive and mixed type inhibition) was calculated from the Michaelis–Menten equation was converted to Lineweaver–Burk equation into a straight line by plotting 1/velocity opposite to 1/[S]. Graphs were plotted by means of GraphPad Prism 7.03 for Windows (GraphPad Software, San Diego, CA, USA, www.graphpad.com, accessed on 13 May 2021) [48].

From obtained equations of regression curves in the Lineweaver-Burk plot, the values of maximum velocity (V_m_) and Michealis constant (K_m_) were calculated. Using non-linear regression (selected panel: enzyme kinetic inhibition/ mixed model inhibition) the values of Ki and alpha were determined. The alpha value defines the mechanism. Its value defines the degree to which the binding of inhibitor changes the affinity of the enzyme for substrate. Its value is always greater than zero. When alpha = 1, the inhibitor does not alter the binding of substrate to the enzyme, and the mixed model is identical with non-competitive inhibition. When the alpha is very large, the binding of the inhibitor prevents the binding of the substrate, and the mixed-model inhibition becomes competitive inhibition, and when alpha is very small (but greater than zero), the binding of the inhibitor enhances substrate binding to the enzyme, and the mixed model becomes nearly identical with an uncompetitive inhibition.

### 3.6. ADME Prediction

Mw, consensus log P, number of hydrogen bond donors and acceptors, rotatable bonds, and topological polar surface area (tPSA), GI absorption and BBB permeation of the quinoxaline derivatives were calculated by the SwissADME software accessed from http://www.swissadme.ch (accessed on 16 May 2021) [49,50].

### 3.7. In Vitro Cytotoxicity Assay

The cytotoxicity assay of the tested compounds against human SH-SY5Y neuroblastoma cells was measured using a sulforhodamine B (SRB) assay [51,52]. The cells were cultured in Dulbecco’s modified Eagle’s medium supplemented with 10% fetal bovine serum (Gibco, Paisley, UK) and grown at 37 °C in a humid atmosphere containing 5% CO_2_. Cells (10,000 cells/well) were seeded in a 96-well plate and incubated at 37 °C, 5% CO_2_ for 24 h. The cells were then treated with or without various concentrations (0, 3.125, 6.25, 12.5, 25, 50, and 100 μM) of each compound for 48 h. After an incubation period, 40% (*w*/*v*) trichloroacetic acid (TCA) was added to the cells, and the cells were then incubated at 4 °C for 1 h. A total of 0.4% (*w*/*v*) SRB solution (100 μL) was added to each well, and the cells were incubated for 1 h at room temperature. The SRB solution was removed, and then the cells were washed three times with 1% (*v*/*v*) acetic acid, and they were allowed to dry at room temperature. The protein-bound dye was dissolved with 10 mM Tris base solution, and the absorbance was measured at 492 nm using a microplate reader.

### 3.8. In Silico Analysis of HuAChE Binding Characteristics

#### 3.8.1. Protein Structure Preparation

The x-ray crystal structure of Recombinant Human Acetylcholinesterase in complex with (−)-huperzine A, PDB ID: 4EY5, was obtained from the RCSB Protein Data Bank (www.rcsb.org, accessed on 20 April 2021), [53] in a PDB format file. Crystallographic water inhibitors were removed, and polar hydrogen atoms were added using AutoDock Tools (ADT; version 1.5.6) [54]. The protein structure was written into a PDBQT format file.

#### 3.8.2. Ligand Preparation

The 3D structure file of the quinoxaline structure used was obtained from the PubChem database. The quinoxaline structure was taken from a structure with CID 7045, and other ligands were created by Public Computational Chemistry Database Project (www.pccdb.org, accessed on 6 April 2021), and saved in mol2 format, which then converted into the PDB format using Obabel [55]. Hydrogen atoms were added to all ligands. Finally, the structure was written into the PDBQT file format using ADT.

#### 3.8.3. Molecular Docking Parameters

Molecular docking studies were performed using the AutoDock4 program, similar to previous studies [56,57]. During the process, the protein structure was set as a rigid molecule with a flexible ligand. The dimensions of the active site box were set at 120 × 120 × 120 cubic angstrom (Å^3^), with the grid spacing of 0.375 Å at the center of the protein structure (x = −2.87, y = −40.07, z = 30.93). The last grid size was considered as the blind docking process. Other parameters followed the default values in ADT. Fifty genetic algorithm (GA) runs with a population size of 200 were performed for conformational sampling. Ligands are arranged by the calculated ΔG value; lower ΔG values correspond to more desirable ligand binding, while higher ΔG values are less desirable [58]. The docking score is the predicted binding affinity in kcal/mol. The calculation of the binding score was previously described [58,59].

To perform the site-specific docking, the PAS site was then defined. The PAS site grid box was based on Trp286. The grid was set at 40 × 40 × 40 Å^3^ with the grid spacing of 0.375 Å from the nitrogen atom the indole ring (x = −19.47, y = −38.04, z = 28.18). The docking protocol was identical to the blind docking process mentioned above.

#### 3.8.4. Molecular Docking Study and Binding Energy Calculation

The molecular docking study was carried out using AutoDock4 program in Ubuntu 18.04. The ligand-protein interaction was visualized and analyzed using the Visual Molecular Dynamic (VMD) package [60]. The hydrogen bond and π–π interaction were considered for the ligand-protein interactions.

## 4. Conclusions

In this work, we presented the design and synthesis of quinoxaline-based compounds as new acetylcholinesterase inhibitors. Drug-likeness and BBB permeabilities were predicted from the synthesized compounds. The potency in acetylcholinesterase inhibition from the synthesized compounds was observed within the same range of known AChE inhibitors. In addition, the peripheral anionic site (PAS) was proposed for the potential binding moiety, leading to the further efficient modification on the quinoxaline scaffold.

## Figures and Tables

**Figure 1 molecules-26-04895-f001:**
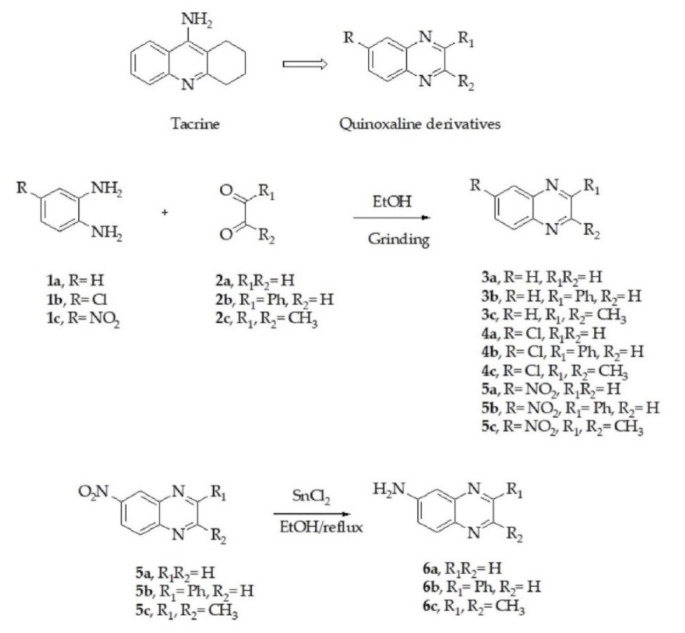
Design strategy and synthesis of quinoxaline derivatives.

**Figure 2 molecules-26-04895-f002:**
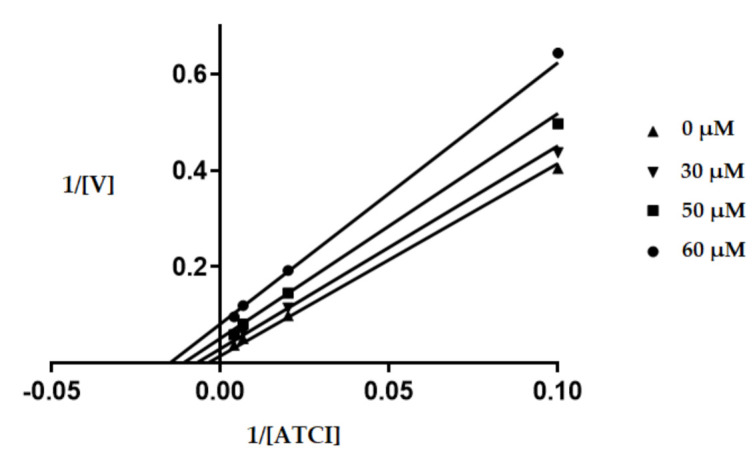
Kinetic assay for AChE inhibition by **6c**. The kinetic assay of AChE inhibition was carried out at 0, 30, 50 and 60 µM of compound **6c**. The concentration of ATCI substrate was varied between 10 to 250 µM. The resulting shown Lineweaver-Burk plot generated by the GraphPad Prism 7.03 software proved that compound **6c** inhibited the enzyme by mixed-type inhibition mode. [ATCI], substrate concentration (µM); V, reaction velocity (1/V (abs/min)^−1^).

**Figure 3 molecules-26-04895-f003:**
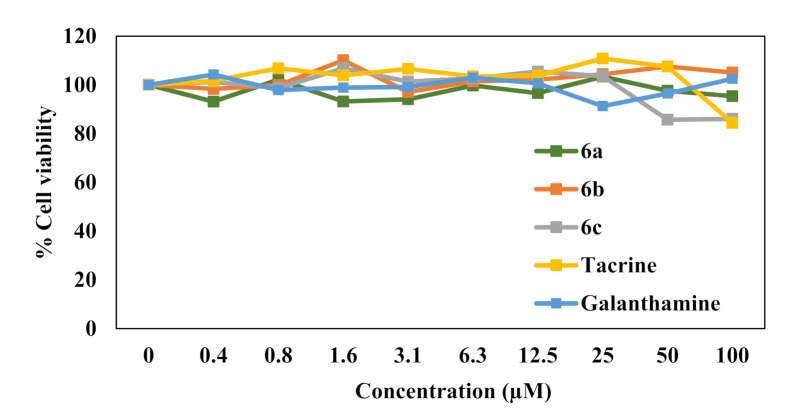
The cytotoxic effect of synthesized compounds on SH-SY5Y cells as determined by SRB assay. Cells were treated with the synthesized compounds at different concentrations for 48 h, and then SRB assay was performed. Data are means  ±  SD of three independent experiments, and %cell viability was calculated relative to nontreated control.

**Figure 4 molecules-26-04895-f004:**
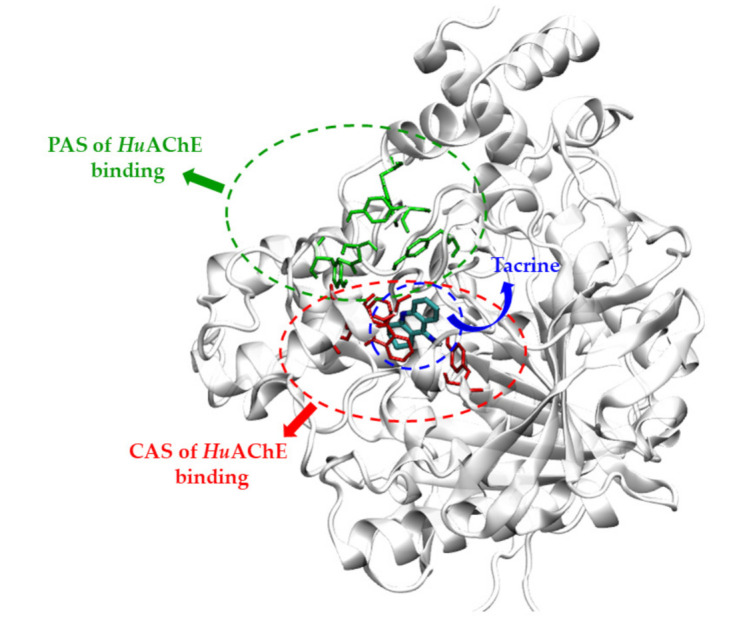
PAS and CAS regions of *Hu*AChE enzyme along with the structure of tacrine.

**Figure 5 molecules-26-04895-f005:**
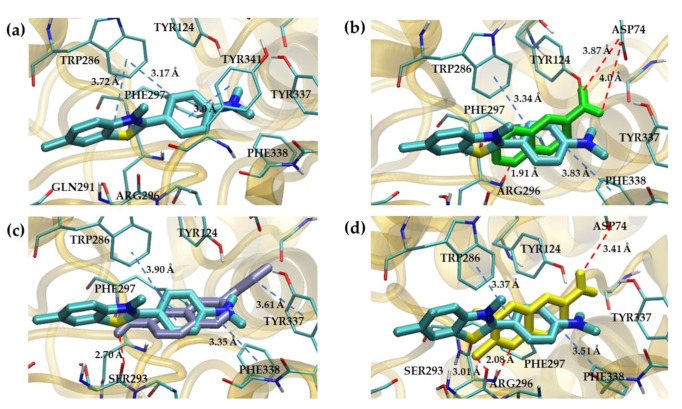
Peripheral Anionic Site (PAS) of *Hu*AChE. (**a**) Three-dimensional docking model of thioflavin T with *Hu*AChE. (**b**) Three-dimensional docking model of compound **6a** with *Hu*AChE. (**c**) Three-dimensional docking model of compound **6b** with *Hu*AChE. (**d**) Three-dimensional docking model of compound **6c** with *Hu*AChE. Red and blue lines displayed a hydrogen bond and π–π interaction, respectively.

**Figure 6 molecules-26-04895-f006:**
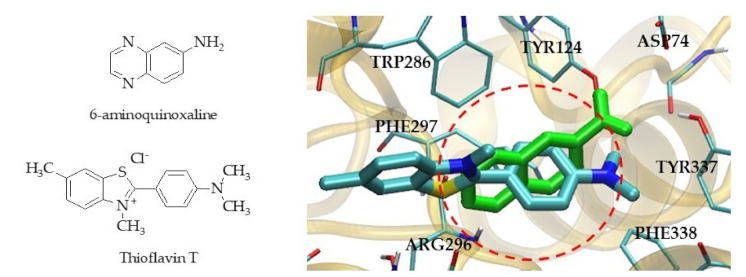
The structure of 6-aminoquinoxaline and thioflavin T showed different surface area between the inhibitor and thioflavin T in PAS site of *Hu*AChE.

**Figure 7 molecules-26-04895-f007:**
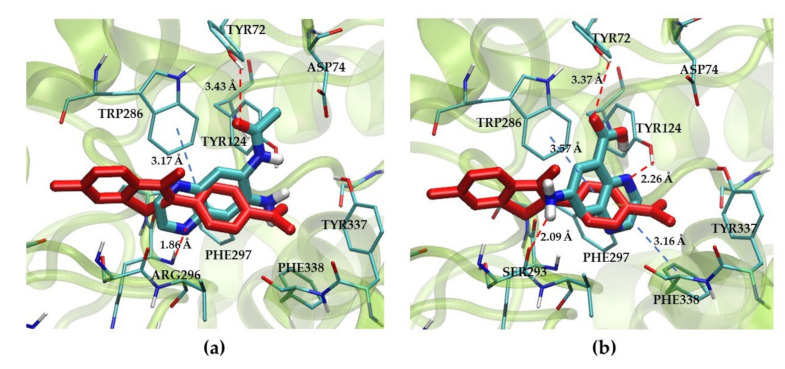
The Peripheral Anionic Site (PAS) of *Hu*AChE, (**a**) 3D docking model of compound **21** complexed with *Hu*AChE aligned by Table 4. (**b**) 3D docking of compound **27** complexed with *Hu*AChE aligned by thioflavin T. Red and blue lines displayed a hydrogen bond and π–π interaction, respectively.

**Table 1 molecules-26-04895-t001:** Physicochemical descriptors and ADME properties of tested compounds calculated by SwissADME.

Code	MW ^a^	Log *P* ^b^	H-Bond Donors	H-Bond Acceptors	tPSA ^c^ (Å^2^)	Rotatable Bonds	GI Absorption	BBB Permeant	Rule of Five ^d^
**3a**	130.15	1.47	0	2	25.78	0	High	Yes	Yes
**3b**	206.24	3.02	0	2	25.78	1	High	Yes	Yes
**3c**	158.20	2.09	0	2	25.78	0	High	Yes	Yes
**4a**	164.59	2.06	0	2	25.78	0	High	Yes	Yes
**4b**	240.69	3.54	0	2	25.78	1	High	Yes	Yes
**4c**	192.64	2.79	0	2	25.78	0	High	Yes	Yes
**5a**	175.14	0.88	0	4	71.60	1	High	Yes	Yes
**5b**	251.24	2.36	0	4	71.60	2	High	Yes	Yes
**5c**	203.20	1.45	0	4	71.60	1	High	Yes	Yes
**6a**	145.16	0.95	1	2	51.80	0	High	Yes	Yes
**6b**	221.26	2.44	1	2	51.80	1	High	Yes	Yes
**6c**	173.21	1.66	1	2	51.80	0	High	Yes	Yes
**Tacrine**	198.26	2.69	1	1	38.91	0	High	Yes	Yes
**Galanthamine**	287.35	1.92	1	4	41.93	1	High	Yes	Yes

^a^ MW: molecular weight, ^b^ log P: Predicted octanol/water partition coefcient log P, ^c^ tPSA: topological polar surface area, ^d^ Rule of five: Number of violations of Lipinski’s rule of five. The selection criteria are as followed: MW < 500, log P < 5, H-Bond donors < 5 and H-Bond acceptors < 10.

**Table 2 molecules-26-04895-t002:** AChE and BuChE inhibitory activity of the quinoxaline derivatives.

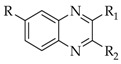						
Code	R	R_1_	R_2_	IC_50_ AChE (µM)	IC_50_ BChE (µM)(%Inhibition)	Selectivity BChE/AChE
**3a**	H	H	H	13.22 ± 4.1	40.64 ± 1.6	3.07
**3b**	H	Ph	H	50.08 ± 2.6	14.91 ± 2.6	0.29
**3c**	H	CH_3_	CH_3_	7.25 ± 1.5	(23.42 ± 1.7%)	-
**4a**	Cl	H	H	23.87 ± 1.3	(37.42 ± 2.3%)	-
**4b**	Cl	Ph	H	28.49 ± 1.6	(41.16 ± 1.5%)	-
**4c**	Cl	CH_3_	CH_3_	10.67 ± 1.4	(37.35 ± 1.3%)	-
**5a**	NO_2_	H	H	21.31 ± 1.5	42.02 ± 1.0	1.97
**5b**	NO_2_	Ph	H	39.0 ± 0.8	60.95 ± 3.4	1.56
**5c**	NO_2_	CH_3_	CH_3_	8.42 ± 1.8	(32.13 ± 1.9%)	-
**6a**	NH_2_	H	H	0.74 ± 0.5	(32.87 ± 2.8%)	-
**6b**	NH_2_	Ph	H	1.31 ± 0.2	(44.14 ± 1.0%)	-
**6c**	NH_2_	CH_3_	CH_3_	0.077 ± 0.01	(29.22 ± 1.95%)	-
Tacrine				0.11 ± 0.01	0.0066 ± 0.001	0.09
Galanthamine				0.59 ± 0.13	11.55 ± 5.5	19.58

IC_50_ values are expressed as the mean ± SD (*n* = three independent experiments). %inhibition at 100 µM are shown in square brackets as the mean ± SD (*n* = three independent experiments). Selectivity index for AChE: IC_50_ BChE/IC_50_ AChE.

**Table 3 molecules-26-04895-t003:** The predicted binding energy of tested compounds **6a**–**6c** at the PAS site.

Compounds	IC_50_ AChE (µM) *Hu*AChE	Predicted Binding Energy in PAS (kcal/mol)
**6a**	0.74 ± 0.5	−5.68
**6b**	1.31 ± 0.2	−7.35
**6c**	0.077 ± 0.01	−6.32
Thioflavin T		−8.29

**Table 4 molecules-26-04895-t004:** The predicted structure design of 6-aminoquinoxaline derivatives (**7**–**36**) by AutoDock Tools 1.5.6.

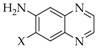	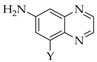
Compounds	X	BindingEnergy (kcal/mol)	Compounds	Y	BindingEnergy (kcal/mol)
**7**	NH_2_	−5.47	**22**	NH_2_	−5.25
**8**	F	−5.39	**23**	F	−5.40
**9**	Br	−6.20	**24**	Br	−6.13
**10**	I	−6.30	**25**	I	−6.25
**11**	Cl	−6.01	**26**	Cl	−5.92
**12**	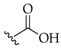	−5.26	**27**	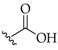	−4.80
**13**	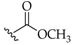	−6.34	**28**	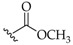	−6.23
**14**	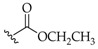	−6.53	**29**	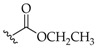	−6.84
**15**	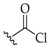	−6.72	**30**	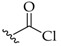	−6.50
**16**	Methyl	−5.94	**31**	Methyl	−5.86
**17**	Ethyl	−6.53	**32**	Ethyl	−6.56
**18**	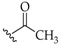	−6.56	**33**	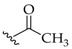	−6.56
**19**	CN	−6.66	**34**	CN	−6.11
**20**	OH	−5.83	**35**	OH	−5.66
**21**	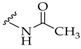	−7.13	**36**	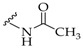	−6.89
Thioflavin T	−8.29	-

**Table 5 molecules-26-04895-t005:** Physicochemical descriptors and ADME properties of guide design new structure of 6-aminoquinoxaline derivatives by SwissADME.

Code	M_W_ ^a^	Log *P* ^b^	H-Bond Donors	H-Bond Acceptors	tPSA ^c^ (Å^2^)	Rotatable Bonds	GI Absorption	BBB Permeant	Rule of Five ^d^
**7**	160.18	0.32	2	2	77.82	0	High	No	Yes
**8**	163.15	1.19	1	3	51.80	0	High	Yes	Yes
**9**	224.06	1.52	1	2	51.80	0	High	Yes	Yes
**10**	271.06	1.57	1	2	51.80	0	High	Yes	Yes
**11**	179.61	1.42	1	2	51.80	0	High	Yes	Yes
**12**	189.17	0.17	2	4	89.10	1	High	No	Yes
**13**	203.20	0.94	1	4	78.10	2	High	No	Yes
**14**	217.22	1.29	1	4	78.10	3	High	No	Yes
**15**	207.62	1.29	1	3	68.87	1	High	Yes	Yes
**16**	159.19	1.20	1	2	51.80	0	High	Yes	Yes
**17**	173.21	1.52	1	2	51.80	1	High	Yes	Yes
**18**	187.20	1.0	1	3	68.87	1	High	Yes	Yes
**19**	170.17	0.76	1	3	75.59	0	High	No	Yes
**20**	161.16	0.30	2	3	72.03	0	High	No	Yes
**21**	202.21	0.43	2	3	80.90	2	High	No	Yes
**22**	160.18	0.48	2	2	77.82	0	High	No	Yes
**23**	163.15	1.16	1	3	51.80	0	High	Yes	Yes
**24**	224.06	1.49	1	2	51.80	0	High	Yes	Yes
**25**	271.06	1.54	1	2	51.80	0	High	Yes	Yes
**26**	179.61	1.13	1	2	51.80	0	High	Yes	Yes
**27**	189.17	0.18	2	4	89.10	1	High	No	Yes
**28**	203.20	0.79	1	4	78.10	2	High	No	Yes
**29**	217.22	1.12	1	4	78.10	3	High	No	Yes
**30**	207.62	1.11	1	3	68.87	1	High	Yes	Yes
**31**	159.19	1.30	1	2	51.80	0	High	Yes	Yes
**32**	173.21	1.19	1	2	51.80	1	High	Yes	Yes
**33**	187.20	0.62	1	3	68.87	1	High	Yes	Yes
**34**	170.17	0.62	1	3	75.59	0	High	No	Yes
**35**	161.16	0.50	2	3	72.03	0	High	No	Yes
**36**	202.21	0.64	2	3	80.90	2	High	No	Yes

^a^ M_w_: molecular weight, ^b^ log P: Predicted octanol/water partition coefcient log P, ^c^ tPSA: topological polar surface area, ^d^ Rule of five: Number of violations of Lipinski’s rule of five (The roles are mol MW < 500, log P < 5, H-Bond donors < 5 and H-Bond acceptors < 10).

## Data Availability

Not applicable.

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
