# Peer review of "Synthesis, Biological Evaluation, and In Silico Studies of New Acetylcholinesterase Inhibitors Based on Quinoxaline Scaffold"

_molecules, 2021, doi:10.3390/molecules26164895_

Round 1
Reviewer 1 Report
The manuscript entitled “Synthesis, Biological Evaluation, and In Silico Studies of New Acetylcholinesterase Inhibitors Based on Quinoxaline Scaffold” deals with 12 newly synthesized AChE inhibitors as a potential treatment for dementia and AD. The manuscript is written well overall, and the research is conducted in a scientifically sound manner. Still, I have some concerns.
In the introduction part, it sounds misleading that AD is the only cause of dementia. It should be discussed carefully and in more detail. Also, the cholinergic hypothesis of AD pathophysiology is not the only one.
Line 59: which flavonoid?
The authors should consider discussing whether a potential AD drug should be a strong inhibitor of AChE. Check this paper: https://pubmed.ncbi.nlm.nih.gov/33999717/ .
Author Response
Response for the reviewer's comment
Q1: In the introduction part, it sounds misleading that AD is the only cause of dementia. It should be discussed carefully and in more detail. Also, the cholinergic hypothesis of AD pathophysiology is not the only one.
Response: We have rewritten the introduction part. We have put the sentence "The dementia can occur in diverse forms including Alzheimer’s disease (AD), vascular dementia, dementia with Lewy bodies and frontotemporal dementia. As Alzheimer’s disease contributes almost 70 percent of the cases, especially in older adults, it is considered as the major form of dementia" (Page2 Line 36-39 ) to respond the comment.
We have also written additional paragraph in the introduction regarding cholinergic hypothesis of AD pathophysiology by the sentences as followings: "Three major hypotheses have been suggested to describe pathogenesis and in turn provide molecular targets for discovery of effective agents for Alzheimer’s disease pharmacotherapy. Histopathological characteristics of AD include extracellular beta-amyloid (Ab) plaques and intracellular aggregations of neurofibrillary tangles (NFTs). Ab plaque resulted from accumulation of insoluble Ab40 fragments, a product of abnormal cleavage of amyloid precursor protein (APP). This leads to beta-amyloid hypothesis for pathogenesis of AD. Tau hypothesis stems from hyperphosphorylation of the microtubule-stabilizing tau protein, which induces tau protein oligomerization and dissociation of tubule subunits. Aggregation of the phosphorylated tau filaments results in neurofibrillary tangles, which are straight, fibrillar, and highly insoluble patches in the neuronal cytoplasm. Amyloid plaques and NFTs play role in the initiation and progression of neuronal damage and subsequently neuronal death [2, 3]. The cholinergic hypothesis is based on selective loss of cholinergic neurons in basal forebrain of AD patients. The deficit of presynaptic acetylcholine production led to cognitive impairment. Administration of inhibitors to acetylcholinesterase improved the symptoms of age-related cholinergic cognitive dysfunction [4, 5]" (Page 2 Line 43-58).
Q2: Line 59: which flavonoid?
Response: We have changed the word "flavonoids" into "flavonoid derivatives" to avoid the reader confusion (Page Line). In addition we also added “flavonolignan” to make them more specific (Page 3 Line 88-89).
Q3: The authors should consider discussing whether a potential AD drug should be a strong inhibitor of AChE. Check this paper: https://pubmed.ncbi.nlm.nih.gov/33999717/ .
Response: We have discussed in more detail about an efficacy and safety of AChE inhibitor in clinical use. We also suggested single AChE inhibitor usage could lead adverse effects such as nausea, vomiting and diarrhea. We also stated about the depression associated with AChE inbhitor as the paper said. We have included it in the citation.
To respond the comment, we have added this sentence “However, clinical use of acetylcholinesterase inhibitors can only temporarily and partially relieve symptoms of Alzheimer’s disease while use of these agents could lead to undesirable side effects such as nausea, vomiting, diarrhea and dizziness. Acetylcholinesterase inhibitors were recently indicated to be prone to increase risk for depression in AD patients [6, 7]. Alzheimer’s disease etiology is multifaceted. Besides the three major hy-potheses, other features such as oxidative stress [8], biometal ions accumulation [9], and neuroinflammation [10] also participate in AD pathogenesis. Therefore, using an agent that can work simultaneously on several targets associated in AD pathogenesis - the mul-ti-target-directed ligands (MTDLs) have been suggested [11-13]” in the introduction section (Page 2 Line 59-67).
Reviewer 2 Report
Authors reported the design, synthesis and biological evaluation of 12 quinoxaline derivatives as acetylcholinesterase inhibitors. Compounds were assayed for their inhibitory activity against human acetylcholinesterase (AchE) and butyrylcholinesterase (BchE) from equine serum. The best compound showed nanomolar inhibition and selectivity against AchE (i.e. 6c, IC50AChE = 77 nM). The most potent BchE inhibitors showed two digits micromolar IC50 values but were not selective. Kinetic studies were performed (6c displayed a mixed-type inhibition) as well as cytotoxicity assays for the three most potent inhibitors (cell viability was not compromised). In silico studies (molecular docking and ADME properties were also performed in silico). Finally, 30 quinoxalin-6-amine derivatives were designed, docked in AchE and their ADME properties predicted.
The manuscript presents important issues/lacks.
-The designed 6-aminoquinoxaline derivatives should be tested and the results included. This is required to support the authors’ design and conclusions from binding mode analyses.
-The three 6-aminoquinoxalines from the first series (6a, 6b, and 6c) were docked in the PAS and in the CAS sites. Best docked poses and their predicted energies of binding were selected and compared with considering one same site and the two different sites. Based on the predicted binding energies, the authors concluded that the three compounds preferred the PAS site. It must be said that 1) generally the best docked pose does not reproduce the experimental binding pose, and cluster analysis or cross-docking are generally preferred/necessary; 2) docking scoring functions have not been developed to compare binding affinities among different binding sites; 3) docking scoring functions are generally not reliable in ranking congeneric compounds. Due to these limitations of the docking method, and the relative close binding energies of 6a, 6b and 6c between the two sites (Table 3) it is not possible to demonstrate by docking that they prefer the PAS site. Also, although the first set of compounds were reported to prefer the PAS site, the 6-aminoquinoxaline derivatives were designed considering the binding mode in the CAS site and their overlap with tacrine. The authors should further clarify this point and explain if this decision was also driven by the fact that 6c was detected as mixed-type inhibitor. Authors reported that docking was assessed by performing a self-redocking of the co-crystallized ligand: unfortunately, this is not sufficient to prove the reliability of the docking method, in this case a retrospective screen is necessary (i.e. enrichment analysis) to check whether the docking protocol is capable to discriminate between actives and decoys (inactives). Moreover, it is reported that a box of 120x120x120 Å was used and centered on the protein structure: the fact that one gigantic box centered on the protein structure was used, suggests that a blind docking approach (over the whole protein structure) was performed instead of a docking targeting the specific pockets (where the box is centered and has a volume based on the binding sites). Authors should clarify this aspect (blind docking should be performed only when the binding site is unknown but never when the pocket is known, as it has poor performances). Authors are invited also to double check the ligand conformations, e.g. in the 3D figures the amino groups are depicted being out of planarity from the quinoxaline ring (e.g. 6a and 6c in Figure 5); the compounds should be flat. The modeling part needs to be improved.
-Dose-response curves of the hit compounds should be added as supplementary data.
-PDB structures of the docked compounds should be provided as supplementary data.
-Please specify the % inhibition instead of n.a. in Table 2
-In line 137-138 it is reported: “...than tacrine (IC 50 =0.107 μM) and galanthamine (IC 50 =0.59 μM), which are clinically used in the pharmacotherapy of AD.” This is not valid for tacrine which has been discontinued. Please fix.
-Docked poses: it would help to have heteroatoms and hydrogens colored by atom type in the 3D ligand structures.
-Missing citations: please add references for the Protein Data Bank and OpenBabel
-Line 477: “...with the grid spacing at the center of the protein structure.” Please clarify/fix this sentence.
-Line 200, Table 3: please specify that the binding energies are predicted binding energies.
-A general double check of the language is required;
-The following typos need to be fixed:
line 26: tarcrine → tacrine
line 69: 7-position → 6-position
line 146: 4c-6c: 3c and 5c should be also included
line 148: Compound → Compounds
line 185: a protein data bank → the Protein Data Bank
line 196: comprised → comprises
line 199: as shown → is shown
line 245: modiffication → modification
line 248: devlopment → development
line 251: ani-line → aniline
line 271: were → are
line 368: Parkinson disease → Parkinson’s disease
line 441: rotat-able → rotatable
Unfortunately the manuscript is still at an early stage, but it has good potential. The above issues need to be addressed.
Author Response
Response for the reviewer's comment
Q1: The designed 6-aminoquinoxaline derivatives should be tested and the results included. This is required to support the authors’ design and conclusions from binding mode analyses.
Response: According to the time and budget limit, we cannot perform the experiment about these design compounds in the current study. In addition, the objective of this study is to suggest the possibility of modified 6-aminoquinoxaline derivatives, which may be for the next study. We are sorry to say that.
Q2: The three 6-aminoquinoxalines from the first series (6a, 6b, and 6c) were docked in the PAS and in the CAS sites. Best docked poses and their predicted energies of binding were selected and compared with considering one same site and the two different sites. Based on the predicted binding energies, the authors concluded that the three compounds preferred the PAS site. It must be said that
Q2.1: 1) generally the best docked pose does not reproduce the experimental binding pose, and cluster analysis or cross-docking are generally preferred/necessary;
Response: Thanks for your comment. We rewrote the manuscript that the docked pose means they preferred the binding site. We have removed all sentences that could lead to misunderstand that the best docked pose can reproduce the experimental binding pose. We only said the comparison of tacrine to an experiment in the sense of implication that “The predicted result seemed reasonable as tacrine was found to bind CAS site of AChE from Tetronarce californica (PDB 1ACJ)” (Page 9 Line 326-328).
Q2.2: 2) docking scoring functions have not been developed to compare binding affinities among different binding sites, and 3) docking scoring functions are generally not reliable in ranking congeneric compounds. Due to these limitations of the docking method, and the relative close binding energies of 6a, 6b and 6c between the two sites (Table 3) it is not possible to demonstrate by docking that they prefer the PAS site.
Response: In the rewritten manuscript, we performed blind docking with the newly optimized structure 6a, 6b and 6c. We found that the compounds clearly preferred the PAS site. Besides, we did not ranking the best compound based on the score. Instead we described the interaction and the role of the functional group in the structure to the PAS site of the protein. The sentence to respond this comment was added in the section 2.4 as “In this study, first of all, we performed molecular docking with the large grid size (120x120x120 Å3), as the blind docking, because the binding site of the synthesized quinoxaline derivative to HuAChE had been unknown. The objective of this process was to roughly guide the possible pocket for the compounds. In Table 3, the compounds 6a, 6b and 6c were selected as representatives for the docking study. The molecular docking predicted that these compounds predictively bound PAS site of HuAChE, Figure 5. As the blind docking suggested that the synthesized compounds bound the PAS site, the PAS site specific docking was again performed to visualize interactions between the compound and the site.” (Page 8 Line 284-292).
Q3: Also, although the first set of compounds were reported to prefer the PAS site, the 6-aminoquinoxaline derivatives were designed considering the binding mode in the CAS site and their overlap with tacrine. The authors should further clarify this point and explain if this decision was also driven by the fact that 6c was detected as mixed-type inhibitor.
Response: We clarified this point of concern. We performed docking experiments again and found that 6a, 6b and 6c preferred only PAS site, not CAS site. Herein this manuscript, we predicted 6a, 6b and 6c preferred the PAS site, not both sites. The first part of docking clearly stated that the synthesized compounds bound PAS site from the sentence “In this study, first of all, we performed molecular docking with the large grid size (120x120x120 Å3), as the blind docking, because the binding site of the synthesized quinoxaline derivative to HuAChE had been unknown. The objective of this process was to roughly guide the possible pocket for the compounds. In Table 3, the compounds 6a, 6b and 6c were selected as representatives for the docking study. The molecular docking predicted that these compounds predictively bound PAS site of HuAChE, Figure 5. As the blind docking suggested that the synthesized compounds bound the PAS site, the PAS site specific docking was again performed to visualize interactions between the compound and the site.” (Page 8 Line 284-292).
Later, we did perform molecular docking at CAS site specifically because we would like to investigate why these compounds did not fit the CAS site like tacrine even though the compounds were at first designed based on tacrine scaffold. The docking at CAS site was only to see the difference of the synthesized compound compared to tacrine in terms of binding. We found that the 6a, 6b and 6c did not fit the CAS site with respect to tacrine. The objective of docking at CAS site was clearly stated to avoid reader confusion from the sentence “As the previously mentioned result showed that quinoxaline derivatives (6a, 6b and 6c) bound PAS site, a question remained unanswer: why these compounds did not fit the CAS site like tacrine even though the compounds were at first designed based on tacrine scaffold. To investigate this question, we applied molecular docking with CAS specific grid box of the compound 6a, 6b and 6c compared with tacrine. The expectation from this approach was to compared the difference in how tacrine and the quinoxaline compounds pose in the CAS site so that the structural modification would be raised from this extracted information.” (Page 9 Line 315-322).
Q4: Authors reported that docking was assessed by performing a self-redocking of the co-crystallized ligand: unfortunately, this is not sufficient to prove the reliability of the docking method, in this case a retrospective screen is necessary (i.e. enrichment analysis) to check whether the docking protocol is capable to discriminate between actives and decoys (inactives).
Response: We agreed with this point. We removed this point in the method as it is not valid. We have replaced the sentence “The reproducibility of the AutoDock4 program was validated via a self-redocking experiment of the native reference ligand.” with “Molecular docking study was carried out using AutoDock4 program in Ubuntu 18.04.” so that it would not misunderstand the reader. (Page 18 Line 629-630).
Q5: Moreover, it is reported that a box of 120x120x120 Å was used and centered on the protein structure: the fact that one gigantic box centered on the protein structure was used, suggests that a blind docking approach (over the whole protein structure) was performed instead of a docking targeting the specific pockets (where the box is centered and has a volume based on the binding sites). Authors should clarify this aspect (blind docking should be performed only when the binding site is unknown but never when the pocket is known, as it has poor performances).
Response: Thanks for suggestion. We clarified this aspect about the blind docking to state that the binding site is unknown. The sentence was rewritten as followings to respond this concern from the reviewer comment: “In this study, first of all, we performed molecular docking with the large grid size (120x120x120 Å3), as the blind docking, because the binding site of the synthesized quinoxaline derivative to HuAChE had been unknown. The objective of this process was to roughly guide the possible pocket for the compounds. In Table 3, the compounds 6a, 6b and 6c were selected as representatives for the docking study. The molecular docking predicted that these compounds predictively bound PAS site of HuAChE, Figure 5. As the blind docking suggested that the synthesized compounds bound the PAS site, the PAS site specific docking was again performed to visualize interactions between the compound and the site.” (Page 8 Line 284-292).
Q6: Authors are invited also to double check the ligand conformations, e.g. in the 3D figures the amino groups are depicted being out of planarity from the quinoxaline ring (e.g. 6a and 6c in Figure 5); the compounds should be flat. The modeling part needs to be improved.
Response: Thanks very much for your suggestion. We did a mistake that the N-C bond between amino group and an aromatic ring need to fixed. Therefore, we redo the molecular docking again with this bond fixed in the whole set of the compounds. The new results indicated the flat geometry as you said. The results were also reanalyzed (Table 2, Table 3, and Table 4) and rewritten corresponding to the newly modeled compounds. Not only Figure 5 but also Figure 6, Figure 7, and Figure 8 were newly illustrated too.
Q7: Dose-response curves of the hit compounds should be added as supplementary data.
Response: Dose-response curves were also included in the supplementary data.
Q8: PDB structures of the docked compounds should be provided as supplementary data.
Response: All structure files of the docked compounds (synthesized and designed quinoxaline derivatives) in PDB format were archived in .zip file and available as supplementary data.
Q9: Please specify the % inhibition instead of n.a. in Table 2
Response: In Table 2, we replaced "n.a" with the %inhibition at 100 µM of the compound.
Q10: In line 137-138 it is reported: “...than tacrine (IC 50 =0.107 μM) and galanthamine (IC 50 =0.59 μM), which are clinically used in the pharmacotherapy of AD.” This is not valid for tacrine which has been discontinued. Please fix.
Response: Thanks very much for your suggestion. We have changed the phrase “which are clinically used in the pharmacotherapy of AD.” into “which are known AChE inhibitors.” to make the statement correct (Page 6 Line 199).
Q11: Docked poses: it would help to have heteroatoms and hydrogens colored by atom type in the 3D ligand structures.
Response: We replace them with the new figures, as suggested, in the manuscript. The new figures are in Figure 5, Figure 6, Figure 7, and Figure 8.
Q12: Missing citations: please add references for the Protein Data Bank and OpenBabel
Response: The citations of both PDB and OpenBabel were included in the ref 50 (Page 17 Line 598-602) and ref 52 (Page 17 Line 605-608), respectively.
Q13: Line 477: “...with the grid spacing at the center of the protein structure.” Please clarify/fix this sentence.
Response: We have clarified this sentence by replacing "...with the grid spacing at the center of the protein structure." with " with the grid spacing of 0.375 Å at the center of the protein structure (x = -2.87, y = -40.07, z = 30.93)." (Page 17 Line 614-615).
Q14: Line 200, Table 3: please specify that the binding energies are predicted binding energies.
Response: We have changed the "binding energy" in Table 3 and Table 4 into "predicted binding energy" to avoid the reader misunderstanding.
Minor issues: Typos
line 26: tarcrine → tacrine
line 69: 7-position → 6-position
line 146: 4c-6c: 3c and 5c should be also included
line 148: Compound → Compounds
line 185: a protein data bank → the Protein Data Bank
line 196: comprised → comprises
line 199: as shown → is shown
line 245: modiffication → modification
line 248: devlopment → development
line 251: ani-line → aniline
line 271: were → are
line 368: Parkinson disease → Parkinson’s disease
line 441: rotat-able → rotatable
Response: We have corrected all these typos.
Round 2
Reviewer 2 Report
Authors' revision and clarifications further confirm that the present manuscript has serious flaws concerning the modeling study and the assumptions behind the molecular design of the 6-aminoquinoxaline derivatives.
1) First of all, it does not sound right to perform a blind docking, affirm that the three hits were predicted to bind the PAS site but later consider the CAS site for designing new 6-aminoquinoxaline derivatives. What's the relevant data for the community from the ligand-protein interactions in the PAS site if also the authors do not consider this one for hit-to-lead optimization? What's the rationale behind assuming that 6a, 6b and 6c bind to the PAS site but not the other quinoxaline derivatives? What the purpose of showing but not considering that docking indicates the PAS site as the "preferred" one?
2) Authors propose a tacrine binding mode where the N group interacts with Glu202 and no clear interaction with the key residue Tyr337. It must be said that in TcAChE (1ACJ) tacrine does not interact with Glu199 (Glu202 in HuAChE, 4EY5) even though a Phe330 (1ACJ) replaces Tyr337 (4EY5). By comparing the structure of the docked tacrine-4EY5 with the experimental tacrine-1ACJ complex, docking misses the expected placement of the tacrine's amino group which experimentally does not interact with Glu202 and the ligand is well packet in a pi-stacking sandwich. Accordingly, the authors have designed new derivatives starting from a binding pose that differs substantially from the available experimental information in a highly similar pocket. Although Phe330 (1ACJ) replaces Tyr337 (4EY5) the binding mode discrepancy is a red flag that requires a deep docking validation. The authors should first demonstrate the robustness of their assumptions by performing for example a docking assessment to check whether their protocol can dock correctly tacrine in 1ACJ. If the docked pose of tacrine in 1ACJ shows a similar pose as the one the authors obtained in HuAChE the main assumption behind the hit-to-lead optimization is not supported. If on the contrary, tacrine docked in 1ACJ well reflects the experimental pose, a retrospective validation (i.e. enrichment analysis) with known binders in 4EY5 is then required. Since the impossibility of performing experimental evaluation of the designed compounds, a rigorous approach for the modeling part is necessary to support the assumptions behind the molecular design.
3) The provided pdb files contain only the ligands, authors should provide the complexes.
Unfortunately, the modeling (docking) part needs to be improved as in the present state does not justify the hit-to-lead optimization approach. The manuscript is still not ready to be accepted in Molecules.
Author Response
Please find the response point by point for Reviewer #2.
Response
Q1) First of all, it does not sound right to perform a blind docking, affirm that the three hits were predicted to bind the PAS site but later consider the CAS site for designing new 6-aminoquinoxaline derivatives.
Q1.1) What's the relevant data for the community from the ligand-protein interactions in the PAS site if also the authors do not consider this one for hit-to-lead optimization?
Response: We do agree with the reviewer after we re-read the manuscript agaim. To make it clear and reasonable, we have performed the docking based on PAS site and changed the discussion along with hit-to-lead optimization focusing on only PAS site instead. In the study, we used the designed derivatives to study and investigated the interaction compared with PAS-bound ligand (thioflavinT). The PAS site remained of interest as it could have the potential to obstruct AChE-induced beta amyloid aggregation. We have also provide the information regarding the importance of PAS site in terms of drug-binding in the sentence “The mentioned result showed that quinoxaline derivatives (6a, 6b and 6c) bound PAS site. This finding complied with the enzyme kinetic study that suggested 6c as a mixed-type inhibitor which approached uncompetitive AChE inhibition. This meant the inhibitor tended to bind with enzyme-substrate complex at another site. We then computationally designed additional quinoxaline derivatives so that the binding affinity could be improved. Herein, we considered the design guide based on molecular docked pose of the quinoxaoine derivatives (6a, 6b and 6c) with respect to thioflavinT due to the fact that thioflavinT is an AChE ligand bound at PAS site [40, 41]. The blockade at the PAS site can hinder the AChE-induced aggregation of beta amyloid [42, 43]. The difference in structure surface area of the quinoxaline compound 6a and thioflavinT (red circle) was visualized in Figure 6. The empty space would thus further explored an opportunity for the structure modification by replacing/adding different substituents, which could provide more interactions through hydrogen bonding with the polar nearby amino acids at the PAS site.” (Page 9 Line 334-346).
Q1.2) What's the rationale behind assuming that 6a, 6b and 6c bind to the PAS site but not the other quinoxaline derivatives?
Response: 6a, 6b and 6c were selected for molecular modeling at PAS site because they are the most potent compounds in the series. We also added the sentence “As the compounds 6a, 6b and 6c exhibited the AChE potency, these compounds were selected to investigate more details in compound binding with AChE.” (Page 7 Line 259-260).
Q1.3) What the purpose of showing but not considering that docking indicates the PAS site as the "preferred" one?
Response: We have performed the docking based on PAS site according to the molecular docking results towards PAS site. We also have removed all the content of the molecular docking with CAS site to avoid the confusion and create the content consistency.
Q2) Authors propose a tacrine binding mode where the N group interacts with Glu202 and no clear interaction with the key residue Tyr337. It must be said that in TcAChE (1ACJ) tacrine does not interact with Glu199 (Glu202 in HuAChE, 4EY5) even though a Phe330 (1ACJ) replaces Tyr337 (4EY5). By comparing the structure of the docked tacrine-4EY5 with the experimental tacrine-1ACJ complex, docking misses the expected placement of the tacrine's amino group which experimentally does not interact with Glu202 and the ligand is well packet in a pi-stackingsandwich. Accordingly, the authors have designed new derivatives starting from a binding pose that differs substantially from the available experimental information in a highly similar pocket. Although Phe330 (1ACJ) replaces Tyr337 (4EY5) the binding mode discrepancy is a red flag that requires a deep docking validation. The authors should first demonstrate the robustness of their assumptions by performing for example a docking assessment to check whether their protocol can dock correctly tacrine in 1ACJ. If the docked pose of tacrine in 1ACJ shows a similar pose as the one the authors obtained in HuAChE the main assumption behind the hit-to-lead optimization is not supported. If on the contrary, tacrine docked in 1ACJ well reflects the experimental pose, a retrospective validation (i.e. enrichment analysis) with known binders in 4EY5 is then required. Since the impossibility of performing experimental evaluation of the designed compounds, a rigorous approach for the modeling part is necessary to support the assumptions behind the molecular design.
Response: Thanks very much for the suggestion. We do misunderstand before and your comment is very advantageous for designing the CAS site targeting drug in the future. Since in the manuscript has cut the CAS site related content, but discuss only the PAS site, the study design of CAS site would be in the step for another set of the compounds for the further study. Therefore, in this revised manuscript, we have not then discussed this suggested point.
Q3) The provided pdb files contain only the ligands, authors should provide the complexes.
Response: We have provided the PDB files for the ligand-AChE complex as suggested.
